# Effect of Substrates Characteristics on Tribological Behaviors of AlTiN-Based Coated WC–Co Cemented Carbides

Yi Chen [1,*], Li Zhang [2,*], Zhiqiang Zhong [3,*] and Shanlin Wang [4]

[1] Key Laboratory for Microstructural Control of Metallic Materials of Jiangxi Province, Nanchang Hangkong University, Nanchang 330063, China
[2] State Key Laboratory of Powder Metallurgy, Central South University, Changsha 410083, China
[3] Chongyi Zhangyuan Tungsten Co., Ltd., Ganzhou 341300, China
[4] School of Aeronautical Manufacturing Engineering, Nanchang Hangkong University, Nanchang 330063, China
[*] Correspondence: 70793@nchu.edu.cn (Y.C.); zhangli@csu.edu.cn (L.Z.); zyzzq@zy-tungsten.com (Z.Z.)

**Abstract:** The wear resistance of coated tools is a key technical parameter, which is indirectly affected by the substrate phase characteristics. WC–Co cemented carbides with varied WC grain sizes (0.4, 0.7, 1.2 μm) and Co contents (3, 6, 10, 12 wt.%) were used as the substrates. Single-layer $Al_{0.52}Ti_{0.48}N$ and multilayer $Ti_{0.89}Si_{0.11}N/TiAlSiN/Al_{0.52}Ti_{0.48}N$ films were deposited on the substrates by DC magnetron sputtering. Reciprocating friction tests were carried out in the air medium and the 3.5 wt.% NaCl aqueous solution, respectively. In the air medium, the films on the fine and the submicron WC–Co substrates with the weaker carrying capacity became worn through earlier than those on the ultrafine substrates. In the NaCl solution medium, for the ultrafine-grained WC–10Co substrates with different Co contents, the friction coefficients ($F_S$) of the film had a linear negative correlation with the hardness ($H_S$) of the substrates. With the decrease in the WC grain sizes or the Co contents, the wear rates of the coated alloys decreased gradually (by 19.7% and 34.5%, respectively). The regular patterns obtained can provide a reference for the selection and design of the phase composition of the cemented carbide substrates.

**Keywords:** TiSiN/TiAlSiN/AlTiN film; cemented carbide substrate; friction coefficient; wear rate; wear medium

## 1. Introduction

Multi-component and multi-layer metallic nitrides-coated cemented carbides have been widely used in the field of cutting tools to improve the service life [1,2]. TiSiN and TiAlSiN films are typical multi-component films, which can form $Si_3N_4$ amorphous phase in their nitride cubic structure and obtain favorable high-temperature mechanical properties [3,4]. With the improvement of the deposition process and the development and application of various hard films such as multi-component (TiAlSiN, CrAlSiN, AlCrSiWN, etc.), multilayer (TiAlSiN/TiSiN/TiAlN, Ti/TiAlN/TiAlCN, etc.), and nano multilayer composite structure (TiAlN/CrAlN, CrAlSiN/CrAlN, etc.), the hardness (nitride film hardness in mainstream cutting tools >30 GPa) and the strength of the hard films continue to improve. In the classical wear theory, the wear resistance of materials was related to the load, hardness and other factors [5].

For various composite materials (metal matrix or polymer), their internal phase composition, the additive type, and the reinforcement phase (particles or fibers) characteristics (structure, particle size, and percentage content) will directly affect the tensile, the compression, the fracture toughness, the fatigue, the hardness, the wear resistance and other mechanical properties [6–8]. This is also true for the composite structure of the "film + substrate" system. The wear and failure of the coated cemented carbides in the process of use (cutting) are caused by the complex interaction of surface effects (such as the

adhesion, wear, diffusion, and frictional oxidation) and volume effects (such as the crack initiation, scratch, and plastic deformation) [9]. Without the comprehensive investigation of the "film/substrate" system, it is often difficult to obtain the optimum performance of the films. While the hardness and the strength of the film are effectively improved, it is necessary to consider the influence of the substrate (or film-substrate compatibility) on the overall durability of the coated alloys, in addition to the effect of the film characteristics (composition, structure, and process change) [10,11] or the wear medium [12].

Regarding the influence of the physical parameters of the substrate, some authors [13] reported the tribological properties of TiN and TiAlN films deposited on the Cu, HSS steel, and WC–Co cemented carbide. For Cu and HSS substrates, when the elastic modulus ratio of film-to-substrate $E_{co}/E_{su}$ is greater than 1, the wear resistance of the relatively soft TiN film is better than that of hard TiAlN film; for WC–Co substrate, $E_{co}/E_{su}$ is less than 1, and the wear resistance of TiAlN film is better than that of TiN. Another report [14] shows that the substrate can influence the friction corrosion resistance of TiSiN–Ag composite film on its surface in seawater. Compared with the 316L steel and the H65 copper substrates, the TC4 titanium alloy substrate with higher crystal density and higher specific strength has the best tribological performance. As for the pretreatment of the substrate surface, Tillmann et al. [15] reported the influence of three pretreatment methods (nitriding treatment (Nitr.), heat treatment (HT), and heat treatment + nitriding (HT + Nitr.)) on the tribological properties of TiAlN film. Following the HT and HT + Nitr treatment, the friction coefficient of the coated alloy system is lower than that of nitriding treatment only. The wear rate of the TiAlN/HT film-substrate system is relatively high, and this study considered that the reason is the high residual stress in the system. Regarding the influence of the substrate surface hardness on the film wear resistance, the researchers [16] conducted carbonitriding treatment before depositing AlCrN and AlCrTiSiN/AlCrN films on the T10 steel substrate. Under those research conditions, the wear rate of the film was significantly reduced after the increase in the substrate surface hardness. Another study [17] reported that laser texture pretreatment on the substrate surface can significantly improve the anti-adhesive wear performance of TiAlN-coated WC–Co-based cemented carbide.

The above research has described a few influential mechanisms of various substrates on the tribological properties of surface metallic nitride films, but there are few systematic studies on WC–Co cemented carbide (subdivided into different types) substrates. This study focuses on WC–Co-based cemented carbide, which is one of the most commonly used substrates for cutting tools, and systematically studied the influence of the characteristic changes of the WC grain size and the Co content on the overall tribological properties of the coated alloy. Finally, the relationship between the bearing capacity of the film or substrate and the overall wear resistance of the film-based system was discussed.

## 2. Experimental Section

### 2.1. Preparation of Coated Cemented Carbides

The substrates used the typical series of commercial WC–Co-based cemented carbides (follow the National Standards of China, GB/T 18376.1-2008). There are seven types of cemented carbide substrates in this series, including four different WC grain sizes (1.2 μm (S1, M1), 0.8 μm (S2, M2), 0.7 μm (S3, M4), and 0.4 μm (M4~M7), which are determined by Image J software); four different Co contents (3 wt.% (M4), 6 wt.% (M5), 10 wt.% (M6), and 12 wt.% (M7)); and two different systems (WC–Co and WC–Co–(Ti, Ta, Nb)C). The related characteristic parameters are listed in Table 1, including the mass fraction of cobalt, average WC grain size, hardness (HV30), etc.

**Table 1.** Properties of WC–Co cemented carbide substrates and the types of deposited films.

| No. | Film | Substrate | Hardness (HV30/kgf·mm$^{-2}$) | Grain Size (μm) |
|---|---|---|---|---|
| S1 | AlTiN | Fine WC–10Co (Fine 10Co) | 1354 | ~1.2 |
| M1 | Multilayer [1] | | | |
| S2 | AlTiN | Submicron WC–10Co–3.5TiC–9.0TaC–2.7NbC (Submicron 10Co–TiC–TaC) | 1539 | ~0.8 |
| M2 | Multilayer [1] | | | |
| S3 | AlTiN | Submicron WC–10Co (Submicron 10Co) | 1556 | ~0.7 |
| M3 | Multilayer [1] | | | |
| M4 | Multilayer [1] | Ultrafine WC–3Co (Ultrafine 3Co) | 2299 | ~0.4 |
| M5 | Multilayer [1] | Ultrafine WC–6Co (Ultrafine 6Co) | 2020 | ~0.4 |
| M6 | Multilayer [1] | Ultrafine WC–10Co (Ultrafine 10Co) | 1809 | ~0.4 |
| M7 | Multilayer [1] | Ultrafine WC–12Co (Ultrafine 12Co) | 1680 | ~0.4 |

[1] TiSiN/TiAlSiN/AlTiN.

TiSiN/TiAlSiN/AlTiN multilayer PVD films were deposited on cemented carbide substrates by DC magnetron sputtering using CemeCon CC800 deposition equipment. The surfaces of the substrate samples were polished before the film deposition, and then ultrasonically cleaned with acetone solution. Two groups of TiSi targets and TiAl targets are used as sputtering targets, which are arranged at relative positions in the deposition chamber. Then, under the Ar atmosphere of 0.6 Pa pressure, a bias voltage of −100 V was applied to the substrates and sputter-etched for 30 min. Before the film deposition, the chamber was vacuumed to $3 \times 10^{-3}$ Pa, and the substrates were heated to 450 °C. Finally, the substrates rotated and started film deposition under the deposition conditions of 450 °C substrate temperature, −90~−100 V substrate bias voltage, and Ar + N$_2$ mixed atmosphere with a total pressure of 0.6 Pa. TiSiN and AlTiN were deposited separately from the TiSi target and the TiAl target, while TiAlSiN was deposited simultaneously from the two targets.

*2.2. Characterization*

The scanning electron microscopy (SEM, using FEI, Quanta FEG 250, Hillsboro, OR, USA) was used to observe the micro morphologies of the substrate alloys and the wear traces, and the incidental energy dispersive spectroscopy (EDS) modular was used to determine the elemental composition. The operating parameters of SEM correspond to the electron beam voltage (HV) of 10~20 kV and the working distance (WD) of 9.1~11.7 mm. Tribological tests (air medium and 3.5 wt.% NaCl aqueous solution medium) were carried out on the coated alloy samples using the ball-on-flat friction and wear tester (BRUKER, UMT-3, Billerica, MA, USA). The schematic illustration of the friction and wear test process is shown in Figure 1.

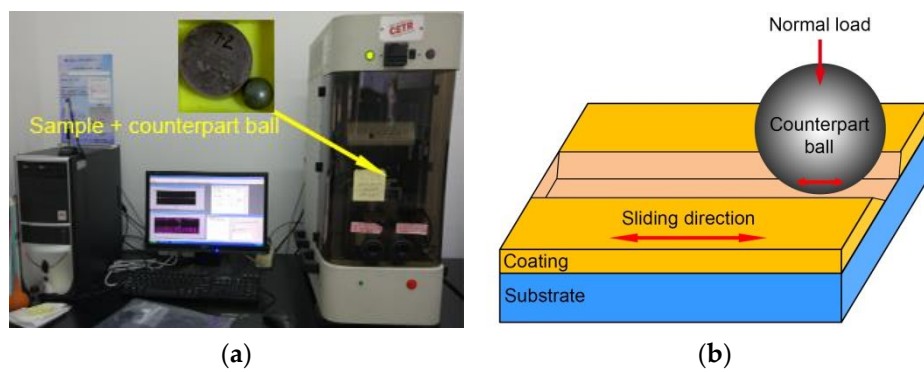

**Figure 1.** Tribological tests device and its working mode: (**a**) UMT-3 equipment and the sample; (**b**) schematic illustration of the ball-on-flat friction.

To ensure sufficient wear depth and avoid wearing out the film at excessive speed (due to the excessive counterpart hardness), the friction counterpart is selected as a $\Phi 6$ mm cemented carbide ball (WC–6Co). Each sample was tested for three grinding cracks. The test parameters were as follows: the test mode reciprocated sliding; the sliding frequency was 5 Hz (corresponds to the sliding speed of 25 mm/s) and the amplitude was 2.5 mm; the normal load was 5 N; the test time was 1800 s, which corresponds to slide 9000 times; and the wear distance was 45 m. The friction coefficient value was directly obtained during the test by the UMT-3 equipment (load range: 5~1000 N, maximum operating frequency: 60 Hz, maximum operating temperature: 1000 °C). A three-dimensional surface profilometer (KLA-Tencor, Alpha-step IQ, Milpitas, CA, USA) was used to measure the cross-sectional area of wear traces after grinding (scanning length $\leq$ 30 mm, vertical measuring range $\leq$ 1.9 mm, scanning speed: 2~200 μm/s), measuring three sections of each wear trace and taking the average value. The wear rates of the coated alloys were calculated by the cross-sectional areas. The 3D surface morphologies of the films were measured by atomic force microscope (AFM, using Veeco, multimode V, Plainview, NY, USA). The test device and the sample are shown in Figure 1a. The schematic illustration of the friction and wear test process is shown in Figure 1b.

## 3. Results and Discussion

### 3.1. Substrate and Film Characteristics

The microstructure and phase characteristics of other WC–Co-based cemented carbide substrates were detailed in the previous literature [18]. The substrate alloy is divided into two series: (1) WC–10Co alloys with different WC grain sizes (M1, M2 and M3) and (2) ultrafine WC–Co alloys with varied Co contents (M3, M4, M5 and M6). Figure 2 shows the cross-sectional SEM images of the single-layer AlTiN (short for "single-layer film") and the multilayer TiSiN/TiAlSiN/AlTiN (short for "multilayer film"). As shown in Figure 2, the thickness of the single-layer film is ~5.4 μm; the overall thickness of the multilayer film is ~3.5 μm (TiSiN 1.9 μm + AlTiN 1.6 μm); and the TiAlSiN interlayer is about 50 nm, which is determined by high-resolution transmission electron microscopy. The design compositions of the two films are $Al_{0.52}Ti_{0.48}N$ and $Ti_{0.94}Si_{0.06}N/TiAlSiN/Al_{0.52}Ti_{0.48}N$ (in atomic fraction), respectively. The characteristics of the film phase and microstructure have been specifically discussed in the previous literature [19]. Figure 3 shows the 3D surface topography of the two films measured by AFM. The surface roughness of the single-layer film and the multilayer film are Ra = 45.8 nm and Ra = 42.3 nm, respectively. That is, the TiSiN layer has a relatively low surface roughness.

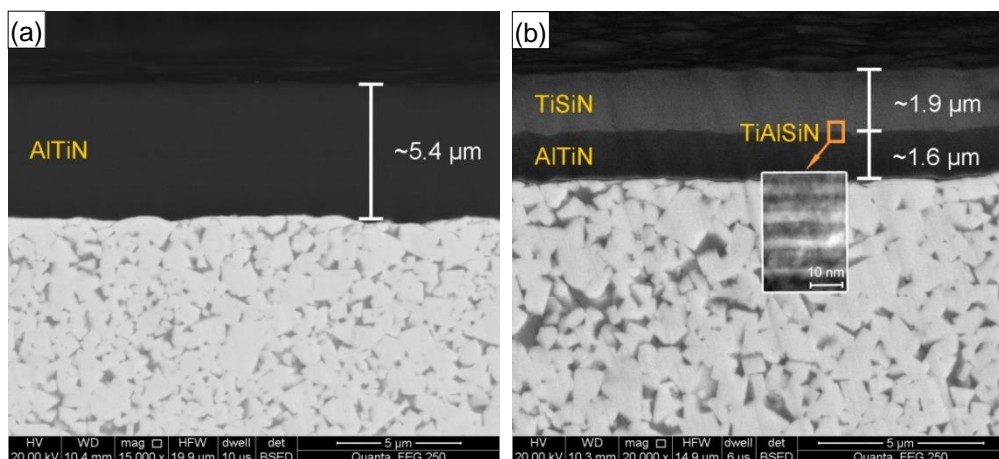

**Figure 2.** Cross-sectional SEM micrographs of AlTiN-based films (submicron 10Co substrates): (**a**) AlTiN single-layer film; (**b**) TiSiN/TiAlSiN/AlTiN multilayer film.

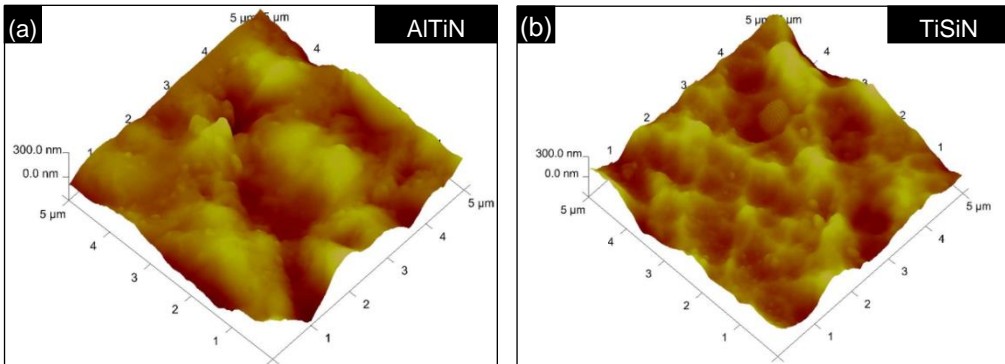

**Figure 3.** Three-dimensional topographic AFM images of the AlTiN-based films (submicron 10Co substrates): (**a**) the AlTiN single-layer; (**b**) the TiSiN surface layer of the multilayer film.

### *3.2. Effect of WC–Co-Based Substrates on Tribological Behavior*

3.2.1. Friction Coefficient and Wear Behavior

Figure 4a,b show the "friction coefficient–time" curves of the coated alloys under the dry friction (air medium) test conditions. Figure 4a shows the results with different WC grain sizes (1.2, 0.7, and 0.4 μm). After the early running-in stage (~180 s), all samples obtained the stable friction coefficient. The curves have two stages of the friction coefficient variation after the running-in. At the first stage (stage I, 180~1400 s), the three coated alloys have similar friction coefficients, and the average friction coefficient is 0.46~0.48. In the second stage (stage II, 1400~1800 s), the friction coefficient of ultrafine 10Co alloy samples remained basically unchanged (0.47), while the friction coefficient of 10Co fine grain and 10Co submicron samples decreased significantly, and the average friction coefficient decreased to 0.40 and 0.41, respectively.

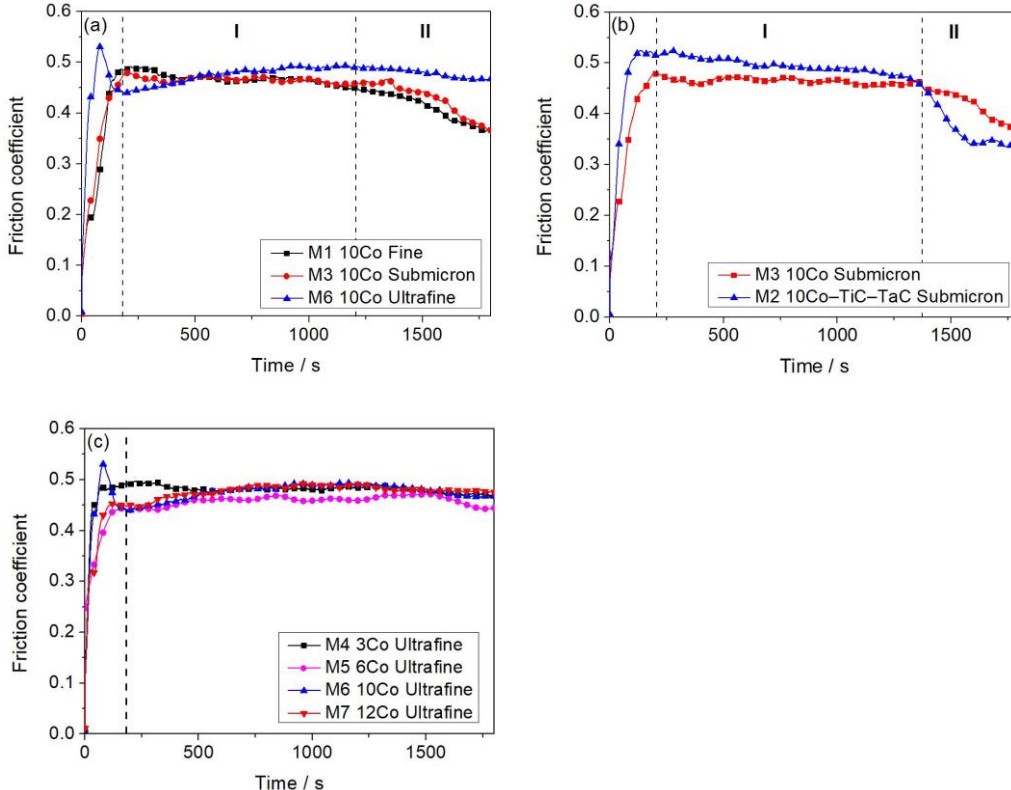

**Figure 4.** Friction coefficients of coated cemented carbides with different substrates: (**a**) 10Co alloys with different WC grain sizes; (**b**) 10Co alloy containing cubic carbide phase; (**c**) ultrafine alloys with different Co contents.

Figure 5a–g shows the SEM photos of the wear trace and the components measured by EDS, which are combined with the friction coefficient curves of Figure 4 for analysis. It can be seen from the EDS results that the surface films of the fine grain (Figure 5a) and the submicron (Figure 5b,d) substrates have been worn through. The components in the middle of the wear trace are the mixtures of the residual films (Ti, Al, N elements), the counterpart ball milling debris (W elements), and the substrates (W, Co elements). The main component is the exposed cemented carbide substrates. As shown in Figure 5c,e–g, the AlTiN layer remains in the middle of the wear trace of the ultrafine substrates, corresponding to the detection of only a small amount of W and Co elements from the counterpart ball. The rough-wear tracks parallel to the sliding direction and the presence of the W and the Co elements in the residual films (from the counterpart ball) indicate that abrasive wear and adhesive wear are the main wear mechanisms of this series of coated alloys [20].

From the above analysis, it can be seen that the change law of friction coefficient corresponds to the different wear processes: (1) Stage I corresponds to the film wear stage. In this stage, the bonding between TiSiN and AlTiN in the multilayer film is good, and there is no friction coefficient fluctuation caused by the peeling between the film layers. At this time, the average friction coefficient is 0.46~0.48, which is the friction coefficient of the multilayer film. In stage I, due to the relatively weak substrate bearing capacity, the counterpart indentation depth of the fine-grained 10Co and the submicron 10Co coated alloys will be greater during the friction process. At this time, the contact area between the counterpart and the film is also larger, and the wear of the films increases. This stage is the main wear stage of the TiSiN layer, and the Si element in the film can provide a certain lubricating effect [21,22]. Compared with the ultrafine 10Co alloy, the wear degree of the fine 10Co and the submicron 10Co alloy gradually increases, and more Si elements enter the wear interface, so their friction coefficients were slightly reduced. Huang et al. [22] studied the friction coefficients of TiAlSiN films under different loads (2, 15, and 25 N)

and the different counterpart balls (stainless steel, $Si_3N_4$, and $ZrO_2$). Under 2 N load, the maximum friction coefficient of the film against the stainless steel ball is 0.84, and the minimum friction coefficient against the $Si_3N_4$ ball is 0.37. With the load increasing to 25 N, the friction coefficient increases in varying degrees, and the increase in the stainless steel ball is the largest (increasing to more than 0.9). $SiO_2 \cdot nH_2O$ lubricating film was generated by $Si_3N_4$ in the TiAlSiN film. Although the load in this paper has not changed, the mechanism is similar. When the bearing capacity of the matrix is weak, the wear of the TiSiN layer is intensified, and the wear-reducing element Si plays a role, resulting in a slight reduction in the friction coefficients. (2) In stage II, for the fine and the submicron coated alloys, the films in the middle of the wear traces are almost completely worn during the reciprocating grinding process, that is, the joint wear stage of the residual films and the substrates. At this time, the friction coefficients mainly reflect the friction characteristics between the counterparts and the substrates. A large amount of wear debris is produced by the grinding between the counterparts and the substrates, and three-body abrasion is formed [23,24]. The three-body debris will isolate the two-body contact between the counterparts and the substrates, reduce the interfacial adhesive material, and thus, reduce the friction coefficients.

As shown in Figure 4b, the friction coefficients of submicron 10Co and submicron 10Co–TiC–TaC coated alloys were compared. In stage I, the average friction coefficients of the two alloys are 0.46 and 0.45, respectively. The friction coefficient drop rate of the submicron 10Co–TiC–TaC alloy with a relatively lower hardness was higher. In stage II (which mainly reflects the wear characteristics of the substrates), the friction coefficient of the submicron 10Co alloy decreases at a higher speed, because the 10Co–TiC–TaC substrate contains more soft phases, which are more prone to wear [25,26]. It can be seen from Figure 5d that the surface film of the 10Co–TiC–TaC substrate has been worn through, and the coated alloy exposed more substrates after the wear test.

Figures 4c and 5e–g show the friction coefficients of the series of ultrafine alloy samples with different Co contents. It can be seen from the figure that the surface films of the ultrafine substrates are not worn through in the whole wear process, and the average friction coefficient is 0.46~0.48. Under the same loading conditions, the films on the ultrafine substrates can play a more protective role. Under the dry friction test condition, the effect of the Co contents on the film friction coefficient is not obvious, and the film friction coefficient of this series of samples is identical (=0.46~0.48).

Figure 6 shows the comparison of friction coefficients with different films. Three stages can be observed: (1) Stage I corresponding to the beginning wear stage, and the average friction coefficients (=0.46, 0.46, and 0.51; average 0.48) of the multilayer coated alloys (M1, M2, and M3) are significantly lower than that of the single-layer film (=0.51, 0.49 and 0.53; average 0.51), with a reduction of 4%~10%. This stage corresponds to the wear of the surface TiSiN layer. The TiSiN layer has a lower surface roughness (Figure 3) and reduces the friction coefficients of the multilayer films effectively. (2) In stage II, with the increase in wear, the surface roughness of the single-layer film gradually decreases, and the friction coefficient of the two films tends to be the same (0.46~0.47). (3) Stage III corresponds to the wear of the substrates and the residual films. Due to more residual films in the wear traces and the larger thickness, the friction coefficient decreasing rate of the single-layer AlTiN in stage III is relatively low.

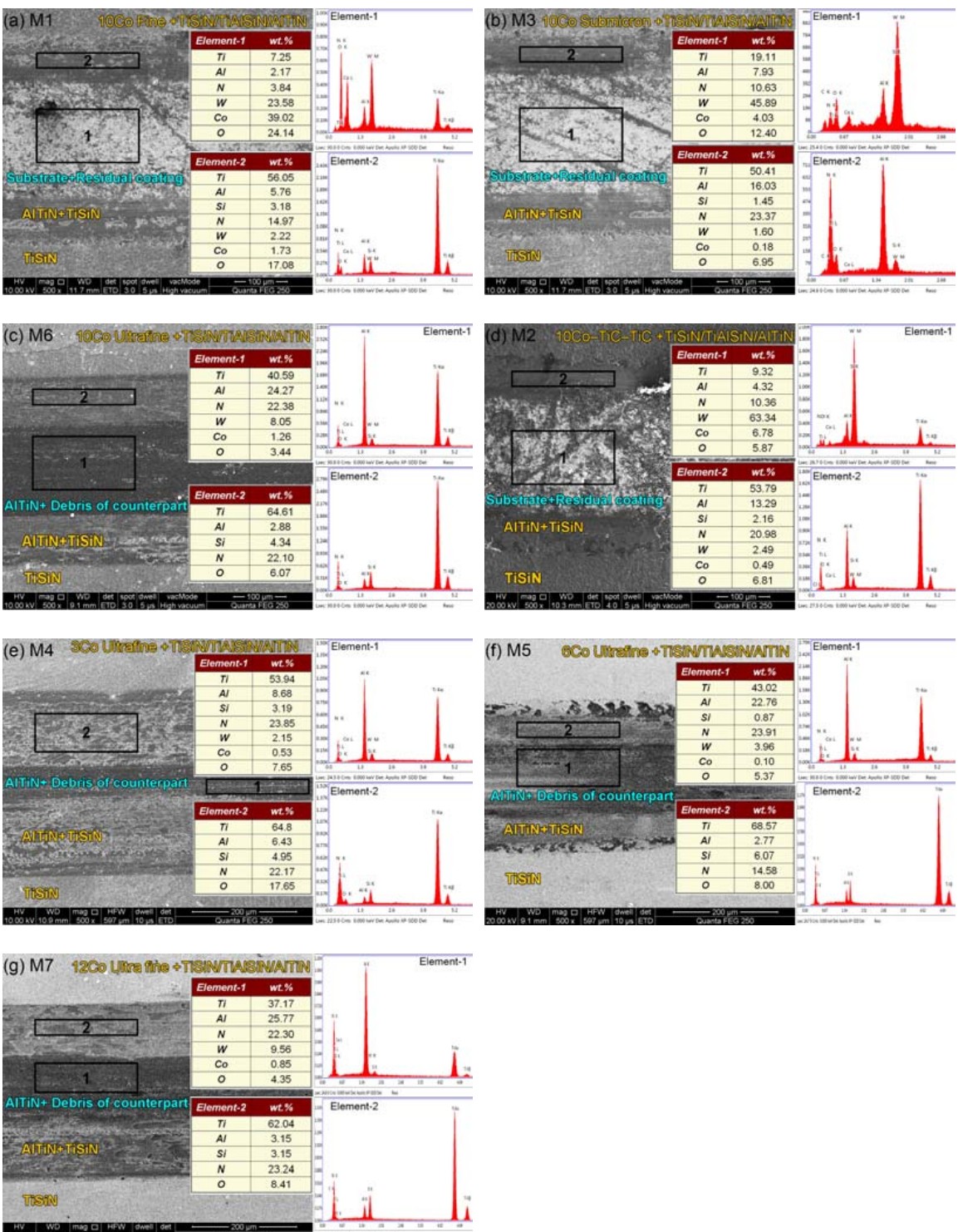

**Figure 5.** SEM images and EDS analysis results of the wear tracks of coated cemented carbides with different substrates: (**a**) M1, (**b**) M3, (**c**) M6, (**d**) M2, (**e**) M4, (**f**) M5, (**g**) M7.

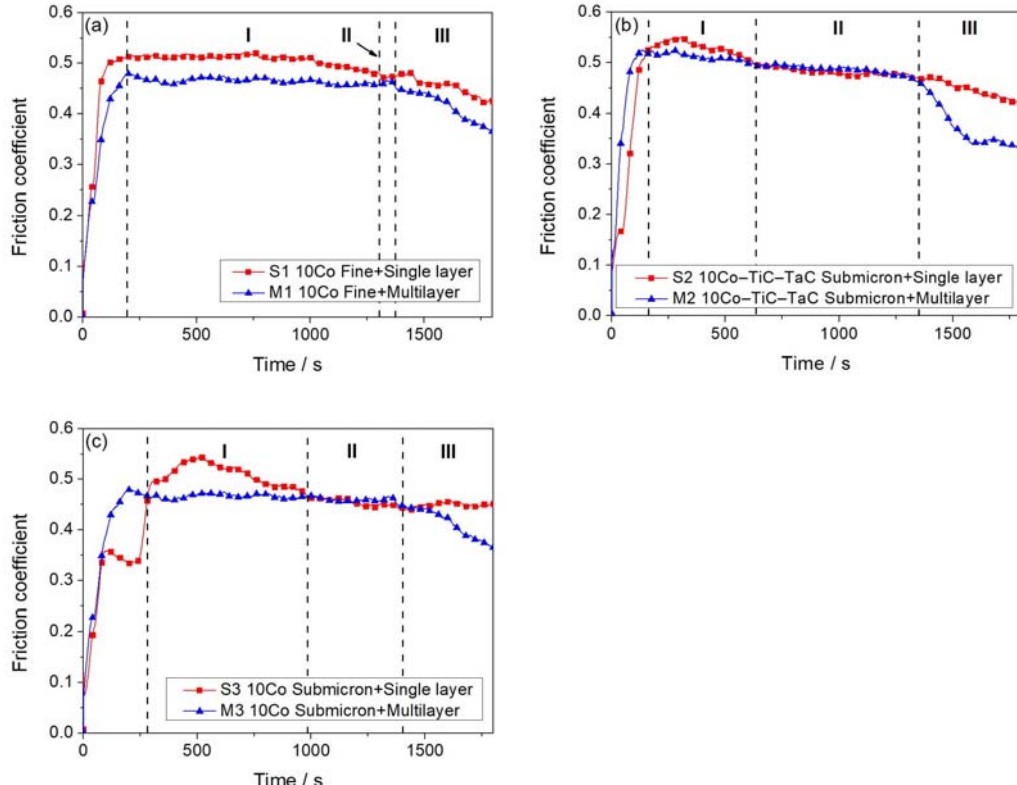

**Figure 6.** Comparison of friction coefficients (corresponding to three wear stages I~III) between the single-layer and multilayer films: (**a**) M1 vs. S1, (**b**) M2 vs. S2, (**c**) M3 vs. S3.

In this study, 3.5 wt.% NaCl aqueous solution was selected as the wear medium, and the tribological properties of coated cemented carbides in a neutral salt solution with certain corrosiveness were compared. The tribological test conditions are consistent with the above dry friction experiments. Figures 7 and 8 show the friction coefficients and the wear morphologies of three groups of typical samples in NaCl aqueous solution medium, respectively. Under the lubrication of NaCl aqueous solution, the generated wear debris is easier to be removed and discharged, and the three-body abrasion caused by the wear debris will also be reduced. Therefore, compared with the dry friction condition, the friction coefficient in NaCl solution is significantly reduced (Figure 7), and the roughness of the wear trace is lower (Figure 8). The average values of the friction coefficients of S3, M3 and M6 samples in the wear stable stage decreased from 0.45, 0.46, and 0.48 to 0.42, 0.35, and 0.3, with the decrease ranges of 7%, 24%, and 38%, respectively. The EDS results show that the W and the Co contents of the film surface (from the counterpart) are very limited, so abrasive wear is its major wear mechanism. The ultrafine coated alloy (M6, Figure 8c)) retains more TiSiN layer, which reduces the amount of wear debris generated during the wear process and corresponds to the smoother friction contact surface. Therefore, the friction coefficient of M6 in NaCl decreases the most. Compared with the combination of "low hardness film + low hardness substrate", the combination of "high hardness film + high hardness substrate" has a more significant anti-wear effect in NaCl aqueous solution medium.

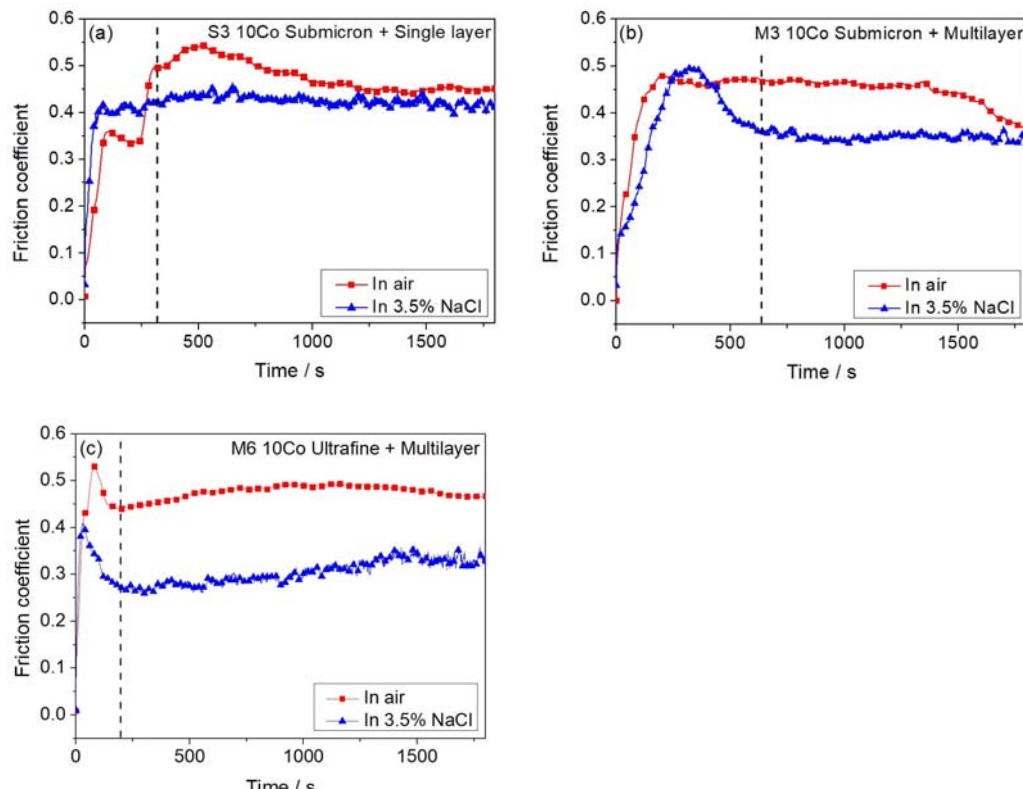

**Figure 7.** Friction coefficient of typical coated cemented carbides in the air and 3.5% NaCl aqueous solution: (**a**) S3, (**b**) M3, (**c**) M6.

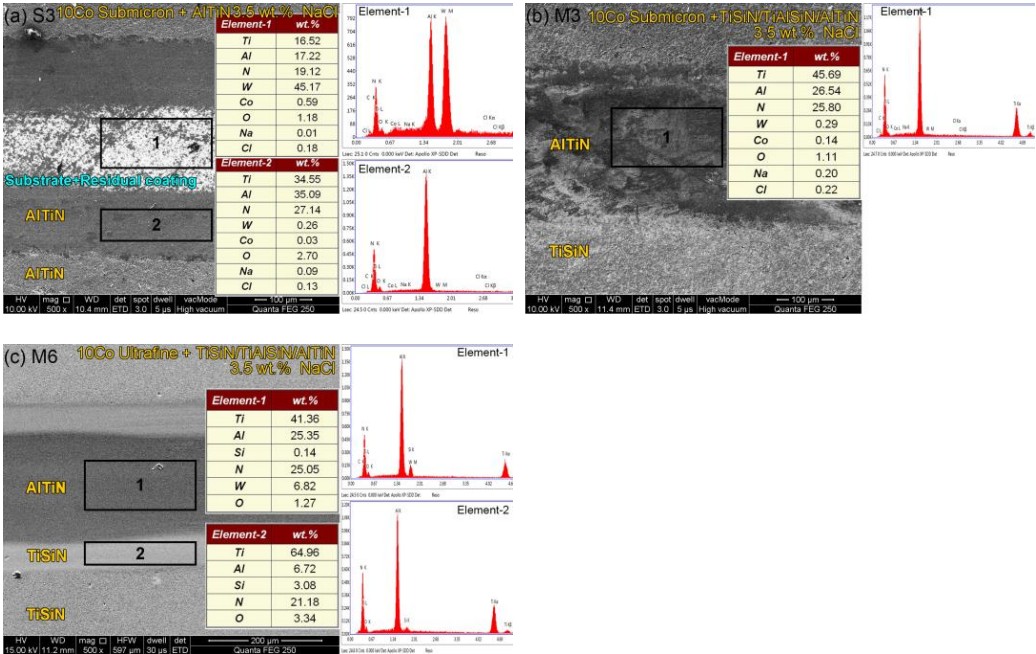

**Figure 8.** SEM images and EDS analysis results of the wear tracks of typical coated cemented carbides in the air and 3.5% NaCl aqueous solution: (**a**) S3, (**b**) M3, (**c**) M6.

As shown in Figure 9a, the film friction coefficient ($F_S$) in NaCl aqueous solution is summarized. The multilayer film still has significantly lower coefficients of friction. There is a negative linear correlation between $F_S$ and substrate hardness ($H_S$), that is, with the increase in substrate hardness, the friction coefficients gradually decrease (from 0.38 to 0.20).

Ultrafine 3Co coated alloy has the lowest friction coefficient (=0.20). Figure 9b shows the corresponding linear fitting results. Under the dry friction condition, the friction coefficients are relatively large due to the intense wear and greater production of wear debris, which makes the above linear influence law hard to reflect. Under the wet friction condition, the wear of coated alloys with lower substrate-bearing capacity is more severe, and the wear degree of the TiSiN layer also decreases with the increase in substrate hardness. The substrates with lower Co contents could retain more TiSiN layers (low surface roughness, containing anti-friction Si element), and their friction coefficients were also lower.

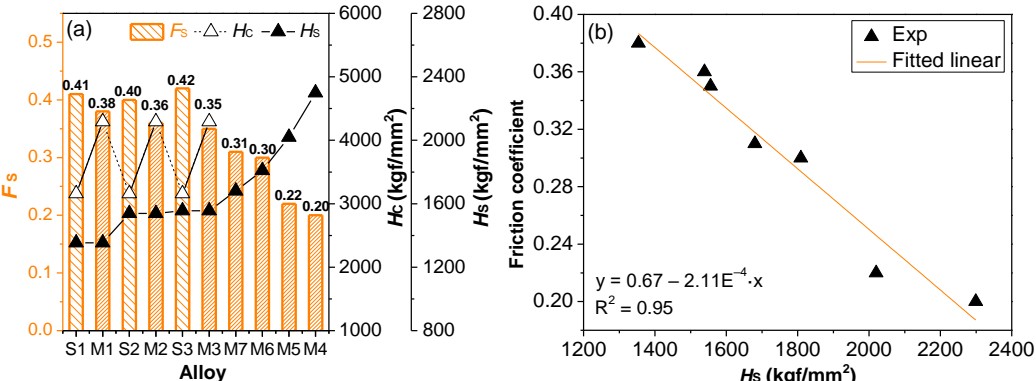

**Figure 9.** Relationships between wear rates $F_S$ of coated cemented carbides in air and $H_C$ (or $H_S$): (**a**) the comparisons of the variation trends between $F_S$, $H_C$, or $H_S$; (**b**) nonlinear fitting of the $F_S$ versus $H_S$.

### 3.2.2. Wear Rate

Wear rate is one of the important data indexes to measure the wear resistance of the films. The wear rates of the coated cemented carbides were calculated by Formula (1) [27].

$$W_s = \frac{V}{F \times L} \tag{1}$$

where $W_S$ is the sliding wear rate (unit: mm$^3$·N$^{-1}$·m$^{-1}$); $V$ is the wear volume of the film (unit: mm$^3$), which is obtained by multiplying the average cross-sectional area of the wear trace by the wear trace length; $F$ is the normal load (unit: N); and $L$ is the total sliding distance (unit: m). The final wear rate is the average value of the calculated results after three repeated tests. Figure 10 is the wear traces' cross-sectional profiles (in air medium) of the coated cemented carbides.

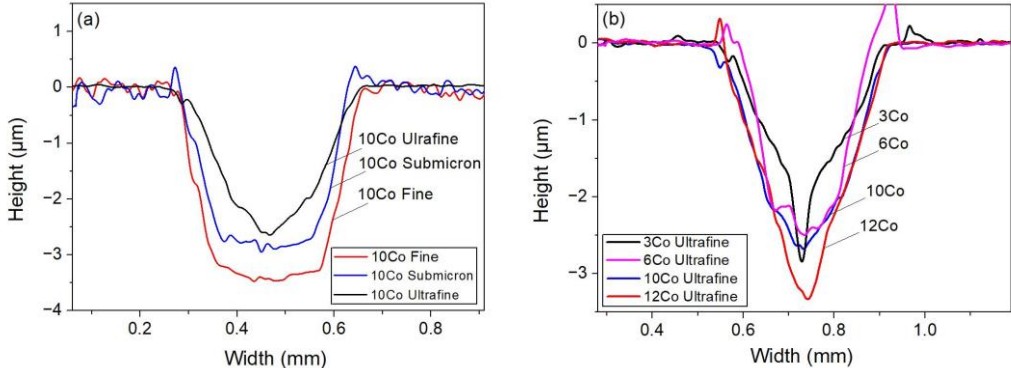

**Figure 10.** Cross-sectional profiles of the wear traces of the coated cemented carbides: (**a**) M1, M3, M6; (**b**) M4, M5, M6, M7.

As shown in Figure 11a, the wear rate ($W_A$) calculation results of the series of coated cemented carbides in the air medium are summarized with the hardness of the substrates

(or the films): (1) When the substrate is the same and the film is different, the multilayer film with high hardness (high plastic deformation resistance) and low friction coefficient has significantly higher wear resistance. Compared with the single-layer coated alloys, the wear rates of multilayer coated alloys are reduced by 33%~46%. (2) When the film is the same, with the decrease in WC grain sizes or Co contents, the wear rates of the coated alloys also gradually decrease, that is, the wear resistance of the coated alloys gradually increases with the increase in substrate hardness. As shown in Figure 11b, the fitting results show that there is a nonlinear exponential negative correlation between the wear rate and the substrate hardness. The cemented carbide substrate with higher hardness can provide the stronger support for the film, reduce the press-in depth of the counterpart, and then reduce the mutual friction contact area; moreover, when the local film is worn through, the cemented carbide substrate with high hardness also has a high wear resistance. In the report [28], compared with the Al 6082 alloy, the Al 6061 alloy with higher hardness had a relatively lower wear volume (4 mm$^3$ vs. 6.2 mm$^3$, 7.7 mm$^3$ vs. 9.8 mm$^3$) in the wear test under different load (10 and 15 N), indicating that the difference of material bearing capacity will affect the wear rate.

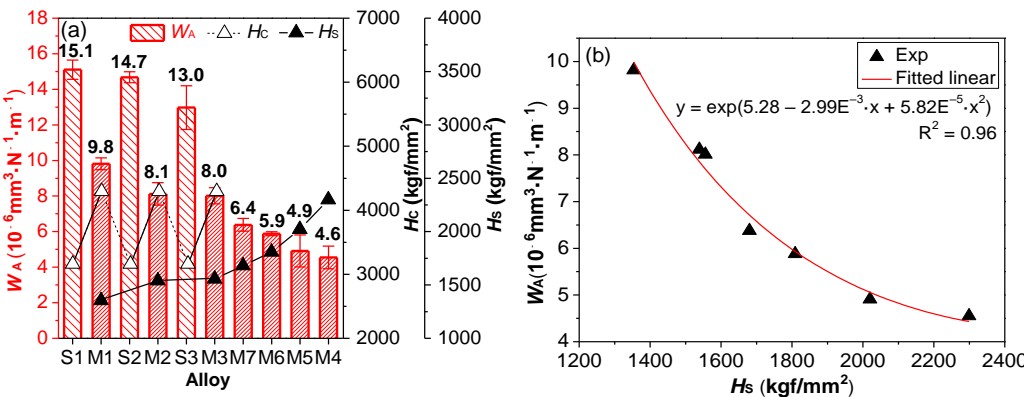

**Figure 11.** Relationships between wear rates $W_A$ of coated cemented carbides in air and $H_C$ (or $H_S$): (**a**) the comparisons of the variation trends between $W_A$, $H_C$ (or $H_S$); (**b**) nonlinear fitting of the $W_A$ versus $H_S$.

Figure 12 shows the wear rate ($W_S$) results of the coated cemented carbides in NaCl aqueous solution medium. It can be seen that the wear rate of each group of coated cemented carbide samples in NaCl aqueous solution medium has decreased by varying degrees. The wear rates of M1, M2, and M3 alloys decreased most significantly, by 28%, 21%, and 23%, respectively. With the decrease in WC grain sizes, the wear rates of the coated alloys decreased by 19.7%, and with the decrease in Co contents, the wear rates of the coated alloys decreased by 34.5%. The surface films of this group of alloys (M2, M2, and M3) were worn through during the dry friction, while remaining intact in NaCl aqueous solution, so the wear resistance was significantly improved. The trend of wear rates in aqueous medium is the same as that in air medium.

In summary, on the premise of obtaining sufficient film adhesion, the bearing capacity of the substrate (or film) is one of the key factors affecting the overall wear resistance of the film–substrate system. For the AlTiN-based coated cemented carbide system composed of two kinds of hard components (hard film + hard substrate relative to steel), increasing the hardness of the film or substrate alone or together can effectively reduce the friction coefficient and wear rate. Compared with the combination of "low hardness film + low hardness substrate" (single-layer film + submicron substrate), "high hardness film + high hardness substrate" (multilayer film + ultrafine substrate) has a more significant anti-wear effect.

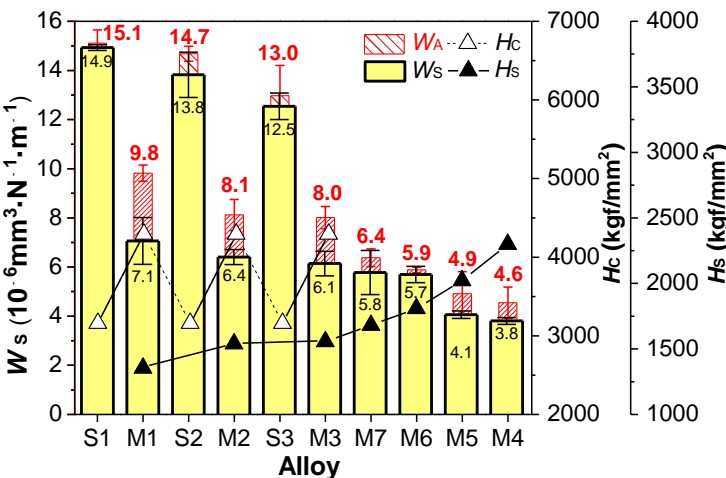

**Figure 12.** Relationships between wear rates $W_A$ of coated cemented carbides in NaCl aqueous solution and $H_C$ (or $H_S$).

## 4. Conclusions

WC–Co cemented carbides with varied WC grain sizes and Co contents were used as substrates. Multilayer TiSiN/TiAlSiN/AlTiN and single-layer AlTiN films were deposited on different substrates by DC magnetron sputtering. The tribological properties were studied by reciprocating friction and wear tests, with a focus on the effect of substrate. The results are summarized as follows:

(1) In the air test medium (dry friction), the average friction coefficient ($F_A$) of the multilayer film is 0.48, and that of the single-layer film is 0.51. Compared with the ultrafine WC–10Co substrate, the friction coefficients of the coated alloys with fine 10Co, submicron 10Co, and submicron 10Co–TiC–TaC substrates decreased significantly in the later stage of the wear process (forming three-body abrasion). The films on the surface of fine and submicron substrates with weaker substrate-carrying capacity were worn through earlier, and the wear rates increased significantly due to the loss of film protection.

(2) In the air test medium, when the film is the same, the wear rates of the coated alloys gradually decrease (wear resistance gradually increases) with the decrease in the substrates' WC grain sizes or the Co contents. There is a nonlinear exponential negative correlation between the wear rate ($W_A$) and substrate hardness ($H_S$). The substrate with higher hardness can provide better support for the film (reduce the penetration depth of the counterpart) and has higher wear resistance itself.

(3) In the 3.5 wt.% NaCl aqueous solution medium, the friction coefficient of the film decreased significantly (0.20~0.42). Under this condition, for a series of ultrafine 10Co substrates with different Co contents, the friction coefficient ($F_S$) of the film is linearly negatively correlated with the hardness ($H_S$) of the substrate, that is, with the increase in the hardness (or the decrease in Co content), the friction coefficient gradually decreases (from 0.38 to 0.20).

(4) In the 3.5 wt.% NaCl aqueous solution medium, compared with dry friction, the wear rate of coated cemented carbide samples decreased to varying degrees. The wear rates of M1, M2, and M3 alloys decreased most significantly, by 28%, 21%, and 23%, respectively. The changing trend of the wear rate is consistent with that in the air medium.

**Author Contributions:** Conceptualization, data curation, resources, and writing—original draft preparation, Y.C.; methodology, writing—review and editing, L.Z. and S.W.; investigation, validation, Z.Z. All authors have read and agreed to the published version of the manuscript.

**Funding:** This research was funded by the Natural Science Foundation of Jiangxi Province, China (No. 20202BABL204032), the Open-end Foundation of Key Laboratory for Microstructural Control of Metallic Materials of Jiangxi Province, China (No. EJ201903432), and the Doctoral Research Initiation Project of Nanchang Hangkong University, China (No. EA201903273).

**Institutional Review Board Statement:** Not applicable.

**Informed Consent Statement:** Not applicable.

**Data Availability Statement:** All data have been shown in this manuscript.

**Acknowledgments:** The authors acknowledge the State Key Laboratory of Powder Metallurgy, Central South University for assisting in characterization.

**Conflicts of Interest:** The authors declare no conflict of interest.

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
