# Peer review of "Effect of Substrates Characteristics on Tribological Behaviors of AlTiN-Based Coated WC–Co Cemented Carbides"

_coatings, doi:10.3390/coatings12101517_

Round 1

Reviewer 1 Report

The paper presents an interesting approach based on the Effect of substrates characteristics on tribological behaviors of AlTiN-based coated WC–Co cemented carbides. However, the innovation of the current research work should be further highlighted and emphasized. At the same time, the authors should consider the following comments to greatly improve the quality of the paper.

1. In the abstract, add a final statement that highlights the importance of this research and its possible potentials. Also, introduce the problem in the initial lines of the abstract.

2. The introduction needs to be improved by relating to the mechanics of the studied materials and their mechanical characteristics. The references to be included are: 10.1177/0021998318790093, 10.1016/j.polymertesting.2017.09.009, 10.1016/j.compstruct.2021.114698, 10.1177/07316844211051733, 10.1002/app.46770, 10.1016/j.porgcoat.2022.107015.

3. Kindly add a table that describes the main physical and chemical properties of the raw materials used in this study.

4. Were the preparation methods described by the authors come in accordance with a certain standard or do they follow previous procedures?

5. What were the operating parameters for the SEM (accelerating voltage and working depth)?

6. The tribological process is not defined properly. Kindly provide all details of the process, including form of wear, wear distance, speed, time and all relevant wear conditions.

7. The conclusion needs to be modified to summarize the research outcomes in short statements with clear observations.

Author Response

Thank you very much for your review, and please see the attachment.

Reviewer 2 Report

The phase characteristics in the alloy substrates will have an important impact on the overall tribological performance of the coated cemented carbides. WC–Co cemented carbides with varied WC grain sizes (0.4, 0.7, 1.2 μm) and Co contents (3, 6, 10, 12 wt.%) were used as the substrates. Single-layer Al0.52Ti0.48N and multilayer Ti0.89Si0.11N/TiAlSiN/Al0.52Ti0.48N films were deposited on the substrates by DC magnetron sputtering. Reciprocating friction tests were carried out in the air medium and the 3.5 wt.% NaCl aqueous solution respectively. In the air medium, the films on the fine and the submicron WC–Co substrates with the weaker carrying capacity are worn through earlier than those on the ultrafine substrates. In the NaCl solution medium, for the ultrafine-grained WC–10Co substrates with different Co contents, the friction coefficients (FS) of the film had a linear negative correlation with the hardness (HS) of the substrates. That is, the friction coefficient gradually decreases with the decrease of Co contents. With the decrease of the WC grain sizes or the Co contents, the wear rates of the coated alloys decreased gradually. There is a nonlinear exponential negative correlation between wear rate (WA) and substrate hardness (HS).

The paper is of original nature, methodology is described correctly. I recommend this paper for publishing with minor revisions.

Specific recommendations:

- measuring ranges and accuracy of measuring devices are not indicated,

- statistical methods - Please provide the statistical test results. Please indicate how many rounds of trials were conducted? Please also report their variances. Please validate the proposed method on multiple datasets.

- the results were not at all compared with the research of other authors. It is difficult to evaluate the correctness of the experiments and the results without comparison. It is necessary to find as close as possible research oriented in terms of materials and parameters, because in the present form it is only the presentation of the results. 

Author Response

(The authors gave the same response as above.)

Reviewer 3 Report

Authors investigated the effect of substrates characteristics on tribological behaviors of AlTiN-based coated WC–Co cemented carbides. This paper is recommended for minor revision.

·     In abstract, …………With the decrease of the WC grain sizes or the Co contents, the wear rates of the coated alloys decreased gradually – please provide how much increases and decreases in properties.

·        TiSiN and TiAlSiN films are typical multi-component films……………………….. Ensure the full form of the abbreviations are at least introduced once in the manuscript.

·        Ensure all the materials name provided properly.  Ex. …………which can form Si3N4 amorphous.

·        Improve introduction with few more literatures.

·        Explain the novelty of the work in the introduction section.

·        What is the standard followed for wear testing?

·        Along with figure 1 provide photo image of the sample and wear setup.

·        Figure 5- improve the font size

·        For the coated alloys with higher substrates hardness or substrates bearing capacity, the depth of counterpart press in films and the wear contact areas are both smaller, and their friction coefficients are also lower. – in many places of discussion part the supporting literatures and citation is missing.

·        Please improve the font size in all the line graphs.

·        ……………………..including four different WC grain sizes (1.2 μm (S1, M1), 0.8 μm (S2, M2), 0.7 μm (S3, M4), and 0.4 μm. How these particle size are obtained? Weather these are average sizes? Please provide particle size analysis.

Author Response

(The authors gave the same response as above.)

Reviewer 4 Report

In the considered manuscript, tribological behaviors of AlTiN-based coated WC–Co cemented carbides were studied. The single-layer Al0.52Ti0.48N and multilayer Ti0.89Si0.11N/TiAlSiN/Al0.52Ti0.48N films were deposited on the substrates by DC magnetron sputtering. The subject matter is very interesting and had a special value considering the practical applications. The paper is rather clearly presented and well organized. The conclusions seem to be sound and justified However, I suggest a mandatory revisions regarding the following points to improve the quality of the paper:

  1)    In the section 3.1. Substrate and film characteristics, the authors wrote: “The characteristics of the film phase and microstructure have been specifically discussed in the previous literature [18].” However, the films described in the paper [18] were produced on the another substrate. Are the authors sure that the phase composition of these films is identical? If so, it is necessary to justify such an opinion.

   2)    The CoF’s profiles vs, time of friction (Figures 4, 6 and 7) are very smooth. Were there no greater fluctuations in the CoF values?

  3)    It seems that the considered manuscript should not be published in a special issue “Laser Surface Modification of Cemented Carbides, Superalloys, Composites, and Alloys” because of the technique used in order to form the coatings. DC magnetron sputtering process is a thin film Physical Vapor Deposition (PVD) coating technique.

Author Response

(The authors gave the same response as above.)
